Ability of clinical data to predict readmission in Child and Adolescent Mental Health Services

Koochakpour Kaban Kaban.koochakpour@ntnu.no 1
Pant Dipendra 1 2
Westbye Odd Sverre 2 3
Røst Thomas Brox 1 4
Leventhal Bennett 5
Koposov Roman 6
Clausen Carolyn 3
Skokauskas Norbert 3
Nytrø Øystein 1 2 7
1 Department of Computer Science, Norwegian University of Science and Technology , Trondheim , Norway
2 Department of Child and Adolescent Psychiatry, Clinic of Mental Health Care, St. Olav University Hospital , Trondheim , Norway
3 Regional Centre for Child and Youth Mental Health and Child Welfare (RKBU Central Norway), Department of Mental Health, Faculty of Medicine and Health Sciences, Norwegian University of Science and Technology , Trondheim , Norway
4 Vivit AS , Trondheim , Norway
5 The University of Chicago , Chicago , IL , United States of America
6 Regional Centre for Child and Youth Mental Health and Child Welfare (RKBU North), UiT The Arctic University of Norway , Tromsø , Norway
7 Department of Computer Science, UiT The Arctic University of Norway , Tromsø , Norway
Elkan Charles
Electronic publication date: 2024 Oct 18
Publication date: 2024
Volume: 10
Electronic Location ID: e2367
Received 2024 Feb 28; Accepted 2024 Sep 7
Copyright: ©2024 Koochakpour et al.
Copyright year: 2024
Copyright holder: Koochakpour et al.
License: This is an open access article distributed under the terms of the Creative Commons Attribution License, which permits unrestricted use, distribution, reproduction and adaptation in any medium and for any purpose provided that it is properly attributed. For attribution, the original author(s), title, publication source (PeerJ Computer Science) and either DOI or URL of the article must be cited.
License URL: https://creativecommons.org/licenses/by/4.0/

Keywords: Machine learning, Prediction, Classification, CAMHS, Clustering, Mental health, Patient readmission

Funding: Norwegian Research Council 269117 Central Norway Regional Health Authority 30233 Norwegian University of Science and Technology (NTNU) The Norwegian Research Council (grant no. 269117) funded IDDEAS project. The Central Norway Regional Health Authority (grant no. 30233) funded data-driven decision aids for child and adolescent mental health in the health platform project, and Norwegian University of Science and Technology (NTNU). The funders had no role in study design, data collection and analysis, decision to publish, or preparation of the manuscript.

==============================
This study addresses the challenge of predicting readmissions in Child and Adolescent Mental Health Services (CAMHS) by analyzing the predictability of readmissions over short, medium, and long term periods. Using health records spanning 35 years, which included 22,643 patients and 30,938 episodes of care, we focused on the episode of care as a central unit, defined as a referral-discharge cycle that incorporates assessments and interventions. Data pre-processing involved handling missing values, normalizing, and transforming data, while resolving issues related to overlapping episodes and correcting registration errors where possible. Readmission prediction was inferred from electronic health records (EHR), as this variable was not directly recorded. A binary classifier distinguished between readmitted and non-readmitted patients, followed by a multi-class classifier to categorize readmissions based on timeframes: short (within 6 months), medium (6 months - 2 years), and long (more than 2 years). Several predictive models were evaluated based on metrics like AUC, F1-score, precision, and recall, and the K-prototype algorithm was employed to explore similarities between episodes through clustering. The optimal binary classifier (Oversampled Gradient Boosting) achieved an AUC of 0.7005, while the multi-class classifier (Oversampled Random Forest) reached an AUC of 0.6368. The K-prototype resulted in three clusters as optimal (SI: 0.256, CI: 4473.64). Despite identifying relationships between care intensity, case complexity, and readmission risk, generalizing these findings proved difficult, partly because clinicians often avoid discharging patients likely to be readmitted. Overall, while this dataset offers insights into patient care and service patterns, predicting readmissions remains challenging, suggesting a need for improved analytical models that consider patient development, disease progression, and intervention effects.

Introduction

Hospital readmission happens when patients are admitted to a health service within a specified time interval after being discharged from an episode of care, with the reason for readmission being related to the initial admission. Readmissions cause stress, inconvenience, and increased costs for families, patients, and the healthcare system (Da Silva et al., 2024). According to the report by the Norwegian Ministry of Health and Care Services (Reggeringen.no, 2015) in Norway, approximately 20% of children and adolescents have mental health issues, and Child and Adolescent Mental Health Services (CAMHS) is responsible for providing them with care. Understanding reasons for discharge and readmissions can help reduce the readmission rate, prevent inappropriate discharges, and ensure continuity of care. Readmission in our study is inferred from the Electronic Health Records (EHR), instead of being a directly recorded feature. The EHR provides a rich source of diverse information. Leveraging EHR data with machine learning (ML) algorithms may help to detect patients who are likely to be readmitted by analyzing their records and grouping them based on their similarities and differences. It is important to interpret ML models to comprehend the complexity, scenarios and outcomes that affect readmission. The incorporation of Artificial Intelligence (AI) into CAMHS offers exciting opportunities for enhancing clinical decision support (CDS). However, it also poses concerns about explainability, trustability, and patient safety related to medical interventions and devices. Nonetheless, AI and ML have the potential to help medical professionals improve the standard of care and support clinical decision-making (Haug & Drazen, 2023).

This study aims to investigate the incorporation of AI into CAMHS, focusing on the predictability of readmission, particularly in terms of clinical utility. To improve the quality of care, we examine clinical decision-making by providing retrospective solutions designed to facilitate comprehension and analysis of complex mental health cases.

Data analysis and AI for patient readmission: a scoping review

In this scoping review, we present recent studies applying data analytics and AI for patient readmission prediction. We focus on the methods, performance, and risk factors of the models. Table 1 compares some studies that used EHR data to predict readmissions for different conditions and time intervals.

Table 1 Literature comparison.

Study	Method	Pros	Cons	Results	
Pakbin et al. (2018)	EHR data for ICU readmissions	High AUROC values, specific time intervals	Limited to ICU readmissions, may not generalize	AUROC: 0.76 (72 h), 0.84 (24 h bounceback)	
Matheny et al. (2021)	Comparison of 5 ML models for 30-day readmission	Emphasized calibration, feasibility of using EHR data	Similar AUROC values, varying calibration levels	0.686 - 0.695 for parametric & 0.686 - 0.704 for nonparametric	
Yu et al. (2015)	Feasibility study for Institution-specific readmission prediction	Flexible, adaptable to context	Requires customization for each institution	Framework with context-aware adaptation	
Golas et al. (2018)	Deep Unified Networks (DUN)	Better prediction than traditional methods	Computationally intensive	Improved 30-day readmission prediction with AUC of DUNs: 0.705 ± 0.015	
Xue et al. (2018)	Logistic regression, functional independence measures	High validation concordance	Specific to rehabilitation inpatients	Validation concordance: 0.85	
Park et al. (2024)	Patient-reported outcome measures for 90-day TJA readmissions	Considered patient-reported outcomes	Focused on 90-day (total joint arthroplasty)TJA readmissions	Readmission rate: 5.8%, AUC, recall, precision >0.5	
De Hond et al. (2023)	ML model retraining and recalibration	Improved AUC with isotonic regression	Focused on short-term (7 days) post-ICU discharge	AUC improved from 0.72 to 0.79	
Da Silva et al. (2024)	Comparison of ML algorithms for 30-day pediatric readmissions	Found best algorithm (XGBoost), high AUC	Pediatric-specific, avoidable readmissions only	AUC: 0.814, Readmission rate: 9.5%	
Zeinalnezhad & Shishehchi (2024)	Data mining, genetic algorithms, SVM	Improved accuracy with genetic algorithms	Diabetic readmissions specific	Accuracy: 73.52%, Readmission rate: 11.4%	
Betts, Kisely & Alati (2020)	Boosted trees model for postpartum psychiatric admissions	Good discrimination and calibration	Specific to postpartum psychiatric admissions	Gradient boosted trees gave AUC = 0.80	
Morel et al. (2020)	XGBoost for mental/substance use disorders	Large dataset, better performance than other models	Specific to mental or substance use disorders and based on claims data	AUC: 0.73	

This study is a part of the Individualized Digital DEcision Assist System (IDDEAS) project, which aims to develop CDS for child and adolescent mental health services. The CDS will focus on preventive care, early diagnostics, intervention, treatment, and case management of psychiatric disorders (Røst et al., 2020). Given the relevance and use of ML and data mining techniques for readmission prediction highlighted in the literature, this paper focuses specifically on CAMHS patients. Our aim is not only to identify and classify possible cases of readmission, but also to challenge the predictability and functionality of readmission prediction in CAMHS.

Data and Materials

The data used in this study consists of aggregated EHR data over 35 years of care provided by the CAMHS clinic at St. Olavs University Hospital, in Norway (Koochakpour et al., 2022), recorded in a domain-specific EHR system, also called BUPdata (Oslo, 2024). The dataset includes structured patient information such as demographics, episodes of care, diagnoses, and treatment (i.e., medication prescriptions). It covers a cohort of 22,643 patients with 30,938 episodes of care, 41,411 referrals (both accepted and rejected), 1,840,045 contacts, 57 units, 22,596 medications, 36,087 regulations, 39,713 prescriptions, and 222,165 diagnoses. Compliance with ethical standards was ensured by anonymizing patient data and obtaining necessary approvals. The age and gender composition of the data includes male and female children and adolescents aged 0 to 18 years, with 52% male, 47% female, and 0.28% ‘gender_0’ (others). Age distribution within the cohort is: 59.2% teenagers, 36.7% middle childhood, and 4% preschoolers. Figure 1 illustrates the relationships between the data components.

Figure 1 The Entity Relationship Diagram (ERD) showing entities and relationships derived from patient information.

Data pre-processing and feature engineering

Data pre-processing and feature engineering involved several steps to ensure the quality and selection of data were suitable for analysis. We have outlined these details in the following subsections.

Episodes of care

An episode of care starts when a patient referral to CAMHS is accepted and ends when the patient case is completed (see Fig. 2). All clinical appointments related to assessment, diagnosis, and treatment, from the point of the referral acceptance to the patient case closure and follow-up, are considered part of the same episode of care (Solheim, 2023). The length of episodes varies, ranging from a few days to several years. An individual patient record may have several episodes of care and each episode has at least one contact (see Fig. 3). The occurrence of the episodes, the duration of each episode, the time passed between consecutive episodes, and the overall sequence of the episodes for each patient are important factors in this study.

Figure 2 Episode of care in CAMHS (Koochakpour et al., 2024a; Koochakpour et al., 2024b).

Figure 3 Patient with multiple episodes of care and contacts.

Episodes of care exclusion criteria.

Episodes of care with assessment as “rejection due to capacity” or “rejection due to professional reasons” or with closing code “rejected” or “did not get started” were not included.

Feature engineering based on episodes of care.

The feature “Tillnextepisode” (Count of days until the occurrence of the next episode) is the target variable for readmission classification. It calculates the readmission period based on the number of days between the end of the current episode and the start of the next future episode for each patient. Episodes of care were characterized by counting associated activities, including outpatient visits, day and 24-hour inpatient stays, administrative and research tasks, therapy, examinations, advisory sessions, and treatment planning contacts.

For the care complexity and level of intensity in each episode, specific features were engineered. The “Length_of_Episode” feature represents the time duration between the start and end of an episode. The “Count_visit” feature is the total number of contacts in one episode. The “Care_intensity” feature is the average number of contacts per day. For measuring intensity per month, the “SD_CareEvent_PerMonth” feature is the standard deviation of the number of contacts per calendar month. The “Num_diagnoses” feature is the number of unique diagnoses per episode. Most episodes (i.e., 9,693) contain only one diagnosis. A total of 56 episodes of care contained six diagnoses and only one with 10 diagnoses. The “Num_medications” feature is the number of unique medications per episode. As with the number of diagnoses, most episodes (i.e., 2,607) had only one prescribed medication. At most, there were seven medications per prescription.

The selection of features has some weaknesses. We chose to use “Length_of_Episode” despite knowing that it is not a representation of severity or complexity of care. While it does not provide insights about the number of contacts or variations between periods (i.e., subintervals), for those with no contacts and with numerous contacts, it is an important feature. “Count_visit”, on the other hand, only reflects the total number of contacts and does not provide information about the timeframe in which the contacts occurred or the distribution of these contacts. Additionally, “Care_intensity” gives us an imprecise average value and different episodes can have the same average. Lastly, “SD_CareEvent_PerMonth” is a measure that represents the variability of care for each episode.

Diagnoses

CAMHS in Norway uses the ICD-10-based multiaxial classification of child and adolescent psychiatric disorders (Directorate for e-health, 2022). The system was originally developed by the World Health Organization (WHO) and has since been adopted for use in CAMHS in Norway (Malt & Braut, 2024). The multiaxial classification has six distinct axes. This research focused on diagnoses coded in axis (1): Clinical psychiatric syndromes, axis (2): Specific disorders of psychological development, axis (3): Mental developmental disabilities, and axis 4: Somatic conditions (see Fig. S1). Axis 5 encodes psychosocial situations, and axis 6 is Children’s Global Assessment Scale (CGAS), neither presenting information about the disorders, so they are excluded from this study. We excluded these axes from the analysis because recording was incomplete and variable during each episode. For instance, only a small percentage of patients had their CGAS scores recorded consistently. Axis 5, the psychosocial situation, was known as a strong predictor among collaborating clinicians, and therefore they wished to give priority to other factors. The distribution of provided diagnoses shows that approximately 73.4% are on axis 1, 8.1% are on axis 2, 1.4% are on axis 3, and 17.2% are on axis 4. For the patients that have been referred to CAMHS for further assessment, any somatic diagnoses provided in other clinics are recorded within the CAMHS system and considered when treating the patient. In our diagnostic codes, we had R-codes (temporary, symptom-based) and Z-codes (used to identify reasons for contact, not otherwise covered within ICD litra A-Y) (Directorate for e-health, 2022). Both were removed as they do not specify disorders. On axis 3, codes 1 - 4, relating to intelligence level, were excluded. Internally used codes like “x-000” (no condition detected) and “x-999” (insufficient information) were also excluded (Directorate for e-health, 2022). ICD-10 codes were mapped to phenotypes using the Phecode system for simpler data analysis. Phecode Map 1.2 (beta) was used, including 9,165 unique ICD-10 codes. Of these, 1,365 were used for diagnoses, and 136 without corresponding Phecodes were assigned manually. The impact of this is studied in later sections. The most frequent diagnosis identified was “F900” (disturbances of activity and attention). In addition, “F321” (moderate depressive episode), “F952” (combined vocal and multiple motortics, Tourette’s syndrome), “F431” (post-traumatic stress disorder), and “F901” (hyperkinetic conduct disorder) are also prevalent in our data set (see Fig. S2).

Prescriptions

The dataset includes CAMHS prescriptions, represented by Anatomical Therapeutic Chemical (ATC) codes, which classify medications into a five-level hierarchy (Norwegian Instituteof Public Health, 2022). In our dataset, the prescription data includes the trade name, ATC code, and ATC name. The number of unique ATC codes varies across levels, with fewer prefixes at higher levels, resulting in less generalization. For example, in our data, level 3 (three-character prefix) has 56 unique codes (e.g., N06B), while level 5 (five-character prefix) has 123 (e.g., N06BA04). In this study, we analyzed only the ATC code, using ATC name and trade name only when ATC codes were unavailable. The ATC code was chosen as it groups medications by use and limits duplicates. Each ATC code is counted once per care episode. The dataset predominantly contains medications from the nervous system (N) group of the ATC code system. We initially removed 11,482 prescriptions without an episode identifier and 4,026 prescriptions without ATC codes linked to four medications brands (Melatonin, Concerta, Metamina, Dexidrine) and five energy drinks. We assigned ATC codes to the medications brands but removed the energy drinks (see Fig. S3).

Other data pre-processing details

As described in a previous study (Koochakpour et al., 2024a), missing or incorrect data are replaced with plausible values or excluded to maintain data quality. This section discusses further data pre-processing.

Episodes

Determining the start and end of the episode was one of the challenges. Referral or entry dates were often inconsistent due to lags, varying paths of referral (e.g., internal from other departments) or errors in recording. To ensure consistency, we used the first and last dates of contact. Contacts with inconsistent date or age information, such as dates before the patient’s birth, were removed (this includes cases with pregnant mothers, and the child enrolled as a patient in a parent episode).

Age

Patient age is calculated at the first contact of each episode. We found 178 episodes with ages above 18, mostly single episodes involving 151 women and 25 men aged 19 to 40. 11 episodes involved individuals over 40, possibly related to expectant parents who were under examination and observation to prevent developmental disorders in their future child. Episodes with negative or zero ages could be due to impending births or recording errors. As we lack complete information for these cases, they were removed. Episodes involving patients with childhood history in CAMHS but aged over 18 are related to the situation that CAMHS clinicians occasionally continue seeing patients they knew from before, even after they turn 18. However, as these cases are exceptions, we removed them from our datasets. Patients aged 18 to 19 were retained.

The patient ages were grouped into intervals based on the children’s developmental stages and the Norwegian school system. To ensure that the defined age groups were clinically meaningful, CAMHS clinicians were consulted. Patients with age intervals of 0–5, 6–11, 12–18 years were designated as ‘preschooler’, ‘middle childhood’, ‘teenager’, respectively.

Gender

The patient’s gender was designated as either female (F) or male (M). This is due to the limitations of the EHR system, which only allows the selection of male or female as gender options. The missing gender values were coded as ‘gender_0’ (others). This ‘gender_0’ category indicates a missing value and does not indicate non-binary/neutral gender.

Codings

Even within the same CAMHS clinic with presumed similar coding practices are assessment and diagnostic practice changes over time as a result of changing guidelines and new coding recommendations (Koochakpour et al., 2024b). Similar patient situations may be described or coded differently over time and needs mapping before analysis. Fortunately, in some cases, consultations with the local CAMHS made the mapping possible; however, some features with coding that had different mapping prior to and post the new code system would have to be excluded. Due to the change of diagnostics guidelines over time (Directorate for e-health, 2022) misuse of codes is possible, such as using the procedure code “Z032” (Observation for suspected mental and behavioral disorders), as an axis 1 diagnostic code. Although “Z032” contributed as one of the most frequently recorded diagnoses on axis 1, it was excluded to avoid inconsistencies. Elsewhere, old internal codes were mapped to clinically equivalent ICD10-codes. The category of a patient contact as “inpatient”, “outpatient”, “inpatient_day” or “inpatient_24hours” was inferred from department type and other indicators (see Fig. S4).

Merging episodes

When analyzing the values in “Tillnextepisode”, some negative and zero values were found. This reflects the existence of three distinct types of episodes: (1) episodes that do not overlap and are not contiguous, (2) adjacent episodes, and (3) episodes that overlap (see Fig. 4).

Figure 4 Types of episodes of care.

As type 2 and type 3 episodes may be unlikely or should not occur, these episodes were merged into a single episode to retain the identity of the first episode. Subsequently, all associated features defining the episode, such as the length of the episode have been recalculated. Merging the episodes resulted in 22,857 episodes of care.

Data cutoff

In Norway, the BUPdata EHR system had been in use for CAMHS for almost 35 years (Koochakpour et al., 2022). Different health regions phased the system out after 2012. In 2018, the Central Norway Regional Health Authority made the decision that St. Olav CAMHS should switch from BUPdata EHR to the general specialist Doculive EHR system. New referrals were documented in Doculive from January 5, 2018 (last new patient date), new consultations of all patients were documented in Doculive from March 5, 2018 (last write date), and by July 3, 2019, all patients had been transferred (last read date). Data shows an unusually large number of discharges in the last six months before January 5, 2018 (see Fig. S5). This date was chosen as a cutoff date for our data.

Episodes with unacceptable length

The longest possible duration of an episode for a patient is 6,934 days, equating to nearly 19 years (see Fig. 5). Three episodes were longer than 6,934 days and were removed from the dataset. Except for those episodes, the rest of the episodes had the following distributions as shown in Fig. 6. The majority of episodes (41%) have a duration of up to one year. Around 29% of the episodes were from 1 to 2.5 years, while the remaining 30% have an episode duration >2.5 years, with the maximum duration of 6,759 days.

Figure 5 Distribution of length of patient episodes of care.

Figure 6 Distribution of episode length ranges in categories.

Re-admission period distribution

The median duration before readmission is 452 days and 75% of the patients were readmitted in less than or equal to 914 days (i.e., 2.5 years). The wide range of values in the last quartile (914–5,109 days) indicates a significant spread of values within this portion of the dataset (see Fig. 7). Patient readmission distribution is shown in Fig. 8. To identify any extreme values that might need to be removed, we used a condition (<= (19 * 365 - 1) - (Patient_age * 365 + Length_of_episode)) to ensure that the episode duration did not exceed the maximum allowable time based on the patient’s age and the length of the episode. This helped us determine if any episodes were unusually long and required further investigation. A readmission period was considered valid as long as it did not start or extend after the patient turned 19 years. All episodes satisfied this condition, so they were all kept.

Figure 7 Distribution of patient readmission periods.

Figure 8 Distribution of patient readmission ranges in categories.

Establishing readmission classes and addressing class imbalance issues

Clinicians expressed interest in knowing if a patient will be readmitted and if so, a readmission range rather than the exact number of days. Therefore, our research focuses on identifying whether a patient will be readmitted or not (i.e., binary classification). For those readmitted, determine the range of readmission (i.e., multi-class classification) as short (approximately 0 - 6 months), medium (approximately 6 months–2 years), and long (approximately over 2 years). 15,1% of admissions were readmissions , while episodes without readmission make up 84.9% of the data. This can cause an imbalance issue in binary classification, so it was necessary to establish readmission status classes (see Table 2). For selecting readmission multi-classes (short, medium, long), two factors were taken into account: 1. the significance of clinically relevant readmission periods, and 2. not having very imbalanced data. Finally, after pre-processing, the dataset contained 22,676 episodes of care labeled as in Table 2 below.

Table 2 Distribution of readmission status in classes.

Classifier type	Readmission class	Count of episodes of care within the class (a)	Count of episodes of care out of the class (b)	Total	Class imbalance ratio (ratio of each class size in comparison to the total of the rest) i.e., = (a /total) * 100	
Binary	Not-readmitted	19,250	3,426	22,676	84.9%	
Readmitted	3,426	19,250	22,676	15.1%	
Multi-class	Readmitted in short period (0–6 months)	1,001	2,425	3,426	29.2%	
Readmitted in medium period (6 months - 2 years)	1,316	2,110	3,426	38.4%	
Readmitted in long period (over 2 years)	1,108	2,318	3,426	32.3%	

Methodology

The IDDEAS project, of which this study is a part, was assessed by the Regional Committee for Medical and Health Research Ethics (REK) under reference number 15600, South East. REK concluded that “the project falls outside the scope of the Health Research Act, cf. ‘Data and Materials’, and can therefore be implemented without REK approval” (reference number: 2018/2186, 09/10/2019). Following REK’s recommendation, the IDDEAS research team submitted a risk analysis to the Central Norway Regional Health Authority and a Data Protection Impact Assessment (DPIA) to the local data protection officer, and approval was received.

Figure 9 outlines our study methodology. We describe the methods we applied for dimensionality reduction in data preparation, and how we utilized both classification (supervised) and clustering (unsupervised) methods for our analysis.

Figure 9 Illustration of the overall methodology.

Dimensionality reduction and feature selection

Before classification and clustering, we applied dimensionality reduction and feature selection to remove highly correlated features and assess their correlation with the “Tillnextepisode” target variable. We used principal component analysis (PCA) to identify the components that explained most of the variance in the data. We also used correlation analysis to assess the relationships between the features and the target variable, ensuring that the relevant features were retained for further analysis. These methods may be useful in enhancing the model’s performance by reducing the number of features in the dataset.

Classification

The initial approach was to classify the entire dataset into five classes, one of which represented not-readmitted cases, and the remaining four represented various readmission periods. Despite the high F1-score of the classification model, it had difficulty predicting classes other than the not-readmitted class. This is a data imbalance issue (Lemaître, Nogueira & Aridas, 2017). To address this, we changed the approach and simplified the classification task by cascading it into binary (predicting readmitted or not) and multi-class (predicting readmission period if readmitted). This approach enabled us to manage the imbalance issue more effectively and select the most appropriate classifier for each type.

It appeared that the dataset for the binary classifier could suffer from significant class imbalance (not readmitted: 19,250, readmitted: 3,426). Meanwhile, the dataset for the multi-class classifier could potentially face more severe problems due to its small size (just 3,426 readmitted episodes), rather than class imbalance. To address these issues, we implemented class weighting and naive random oversampling (Lemaître, Nogueira & Aridas, 2017). Furthermore, we reduced the number of classes representing readmission periods from four to three.

We relied on prior insights (Koochakpour et al., 2024b) to select logistic regression due to its simplicity and clear interpretability. Decision tree and random forest were utilized for their abilities to handle non-linear relationships and high-dimensional datasets, respectively. Gradient boosting and XGBoost were included for their high performance and scalability, with XGBoost excelling in managing imbalanced data, especially for our binary classification needs. Multilayer perceptron was tested for its deep learning capability to detect complex patterns. We tried to harmonize the need for interpretability with performance, which is important for understanding predictive factors.

We used cross-validation to assess the performance of our models across different subsets of the data, which helped to prevent overfitting and ensured that the model generalized to unseen data. Specifically, we utilized a 5-fold cross-validation technique using 80% of the dataset for the initial training. For the final model evaluation, we reserved 20% of the entire dataset as an independent testing set. To address data imbalance and small-size dataset issues, we applied and compared class weighting and oversampling techniques. Class weighting adjusts the weight of each class inversely proportional to its frequency, and oversampling involves duplicating samples from the minority class to balance the dataset. These techniques were important in enhancing the model’s ability to predict the minority class. Therefore, we applied this solution in three scenarios: (1). Without class weight or oversampling, (2). With custom/balance class weight, (3). With oversampling. Finally, we evaluated the performance of classifiers to select the best-performing models.

Clustering

Clustering was chosen as one of our methods to explore and understand intra- and inter -cluster distance among episodes of care. Clustering utilized all features and labels from the classifiers. The clusters are compared to the readmission classifiers ‘T illnextepisode’ label. We identified the optimal cluster number and discussed its relation to the classifier classes. We also examined data patterns and distributions in other features and their connection to ‘T illnextepisode’ within these clusters. Having mixed data types (categorical and numerical) in the dataset, the K-prototype algorithm was chosen for clustering (Huang, 1998). The k-modes library (De Vos Nelis, 2015-2021) was used to implement the K-prototype algorithm. We analyzed how different combinations of data affected the clustering outcomes. We used datasets with and without diagnoses and medications. Out of the 1,427 unique diagnoses and medications that were added as one-hot encoded columns in the dataset, 1,305 columns corresponded to ICD diagnosis codes and 123 columns corresponded to ATC codes. Additionally, we evaluated the 20, 50, and 100 most frequent codes under the assumption that they might cover most of the diagnoses and medications in use. For each dataset combination, we computed the optimal number and evaluated the quality of clusters.

Evaluation

The performance of these classifiers was evaluated on the independent test (reserved 20%) set across both binary and multi-class tasks using metrics AUC, F1-score, precision, recall, and accuracy. These metrics were chosen based on their relevance and suitability for addressing the specific challenges and objectives of our research: Precision and recall are important in evaluating the performance of our imbalanced models. The F1-score combines precision and recall into a single measure, providing a balanced evaluation of the model’s performance. Recall, precision and F1 are particularly useful when the cost of false positives and false negatives is high, as is the case in healthcare readmission prediction. We evaluated the model’s sensitivity (true positive rate) and false positive rate using the ROC curve, and AUC for comparing the overall performance of models. The higher the AUC score, the better the model performs across all classes. Finally, the best-performing models were selected as the main result of the paper. Classification outcomes were visualized using 3D scatter plots to compare predictions against actual target variables.

The number and the quality of clusters were analyzed using elbow plot, and Silhouette (SI) and Calinski-Harabasz (CI) scores respectively. SI score analyzes the similarity of data points within a cluster to data points in various other clusters, with a range of −1 (poor clustering) to +1 (perfect clustering). CI score measures the compactness and isolation of clusters, with higher values indicating better clusters. We analyzed the clusters using minima, maxima, and box-plots. To evaluate the process and confirm the clinical relevance of all results, we presented them to a group of clinicians for interpretation and assessment. To ensure rigour in methodology, we followed the medical informatics ML checklists by Cabitza & Campagner (2021) and Cerdá-Alberich et al. (2023) guiding the data and ML analytic, selection of metrics, evaluation and validation processes, and ensuring adherence to ethical and technical standards in healthcare predictive modelling.

Results

Dimensionality reduction and feature selection results

Principal component analysis for dimensionality reduction

PCA reduces the dimensionality while retaining as much information as possible. The PCA is implemented using “sklearn.decomposition” module identified that 12 out of 16 initial features are required to cover 95% of the data variance (see Fig. S6).

Correlation analysis for feature selection

Correlation analysis was used to determine the correlation of features to the target variable and to select the predictors most correlated to the outcome/target label. Scatter plots revealed no significant linear relationships or normality between certain features and “Tillnextepisode” (see Fig. S7). Therefore, among Pearson, Spearman, or Kendall, the Kendall correlation was used to construct the correlation matrix, as it better handles non-linear relationships.

The initial analysis of the correlation matrix showed that some features were either derived, redundant, or complementary (see Fig. S6). As a result, several changes were made to the feature set. From the pairs “outpatient_ratio”—“inpatient_ratio”, and “inpatient_daynight_ratio”—“inpatient_day_ratio”, only one feature was retained in each pair. The feature “Care_intensity” was removed due to its high correlation with both “Count_visit” and “Length_of_Episode”. “Examination_ratio” was also removed because it showed a correlation with “Therapy_ratio” and not a strong correlation with “Tillnextepisode”. It is worth noting that the correlation between “Tillnextepisode” and “label” (readmission class) in the correlation matrix reveals that categorizing “Tillnextepisode” into classes could lead to a loss of information of about 11%. After these adjustments, a new correlation matrix of the final 12 features was generated (see Fig. 10).

Figure 10 Correlation matrix showing the correlation between features and target variable “Tillnextepisode”.

Figure 10 shows “Age_group” and “Inpatient_day_ratio” have the strongest correlation with “Tillnextepisode”. Conversely, the activity ratios (“Therapy_Ratio”, “TreatmentPlanning_Ratio”, and “Counseling_Ratio”) have the lowest correlation. In addition to the literature in Table 1 and Koochakpour et al. (2024b), we used domain-experts for the causality analysis.

Features and target variables

Tables 3 and 4 present the final features and target variable for the readmission classification and clustering tasks. The dataset contains numeric, categorical, and string (or object) data types. One-hot encoding was implemented on the “Gender”, “Age_group”, “Diagnoses” and “Medication” features. We indirectly managed missing values by excluding them from our dataset. Specifically, patients without recorded age or any contact (visit) were omitted. Missing gender values were categorized as ‘gender_0’ (a separate category). All other features were either calculated or provided, resulting in no additional missing data.

Table 3 Final features.

Feature name	Description of feature	Data type	Range	
Age_group	One-hot encoded feature for values: 0–5 (Preschooler), 6–11 (MiddleChildhood) and 12–18 years (Teenager)	Categorical	0,1	
Gender	One-hot encoded feature for values: F (female), M (male) and gender_0 (others)	Categorical	0,1	
Length_of_Episode	Length of episodes (days)	Numerical	1–6,759	
Count_visit	Count of patient’s visits (contacts)	Numerical	0–4,159	
SD_CareEvent_PerMonth	Standard deviation of the number of patient’s visits per month during the episode	Numerical	0–3.3	
Outpatient_ratio	Ratio of outpatient visits out of total visits (inpatient and outpatient)	Numerical	0–1	
Inpatient_day_ratio	Ratio of inpatient (day) visits out of total inpatient visits (both day and 24 h)	Numerical	0–1	
Therapy_ratio	Ratio of patient’s visits with the activity type of therapy	Numerical	0–1	
TreatmentPlanning_ratio	Ratio of patient’s visits with the activity type of treatment planning	Numerical	0–1	
Advisory_ratio	Ratio of patient’s visits with the activity type of advisory	Numerical	0–1	
Num_diagnoses	Number of diagnoses	Numerical	1–10	
Num_medications	Number of medications	Numerical	0–7	
Diagnoses	ICD diagnosis codes, transformed into one-hot encoding for each unique diagnosis	Object (String)	0,1	
Medications	ATC medication codes, transformed into one-hot encoding for each unique medication	Object (String)	0,1	

Table 4 Target variable.

Name	Description	Data type	Range	
Tillnextepisode	Count of days until the occurrence of the next episode	Numerical	1–5,109	

The “Tillnextepisode” target variable was transformed into classes and bin labels. For classification purposes, it is transformed to the readmission classes: “not-readmitted” and “readmitted” for binary classifiers and “short”, “medium”, and “long” for multi-class classifiers. For clustering purpose and for further comparison of results with actual data, we binned this variable into bin labels (“Tillnextepisode_bins”): “not-readmitted”, “readmitted in 0–182 days”, “readmitted in 182–730 days”, and “readmitted in more than 730 days”.

Classification

As mentioned in Methodology, we decided to have two separate classifiers: one for binary and one for multi-class. We compared six different algorithms: random forest, decision tree, gradient boosting, XGBoost, logistic regression, and multi-layer perceptron. We compared the classifications results with different numbers of diagnoses and medications (the most frequent 20/50/100 and all the diagnoses and medication). The results were largely similar. To validate, we performed McNemar test (Raschka, 2018), which showed significant differences (p < 0.05) between some classifiers, the McNemar test results alone do not guide model selection they serve as an additional check for robustness, especially in multi-class cases. However, because we wanted a representative range of diagnoses and medication data, we decided to keep the most 100 frequent diagnoses and medications along with other features. We substituted the ICD-10 diagnostic codes with Phecodes and the results did not differ significantly. When we used a more general ATC code prefix for medications (ATC level 3 instead of ATC level 5), the results deteriorated in some cases, while in others, there was no change (see Excel S1).

As mentioned before, given our imbalanced data set (specifically for binary classification), we evaluated the algorithms in three scenarios: (1) without change, (2) custom/balance class weight (3) oversampling. We compared the evaluation metrics (AUC, F1-score, recall, precision, accuracy with AUC score prioritized) to select the best models for both binary and multi-class classifiers. In conjunction with handling imbalanced data, we implemented data normalization where appropriate, to accelerate algorithm convergence and reduce bias towards larger values. To avoid potential time-related patterns in the data, we incorporated shuffling in the dataset that was split in 80% (5-fold cross-validation for training) and 20% reserved for the independent test set. Collectively, these strategies significantly enhanced our classification outcomes.

Binary classifier

Binary classifier models trained using different algorithms were compared based on evaluation metrics (see Excel S2). Although some algorithms like random forest and decision tree had higher F1-scores, they did not perform well across all classes and had lower AUC scores. Among all, gradient boosting binary classifier without applying class weight or oversampling, and with class weight gave the highest AUC score (0.7093), but it was biased towards the minority class (did not perform well on both classes) (see Excel S2 for recall, F1-score, and confusion matrix). However, gradient boosting with oversampling (n_estimators =100; see Document S1) achieved the next highest AUC score and performed best overall across all classes on multiple evaluation metrics, so we chose that as our optimal model (see Table 5 below and Fig. S8).

Table 5 Result of optimal binary classifier.

Binary classifier model	Gradient Boosting with oversampeling	
AUC	0.7005	
F1-Score (Test set)	0.59	
Accuracy (Test set)	0.64	
Classes	F1-Score	Recall	Precision	Support	
Not-readmitted (Class 0)	0.64	0.49	0.93	3,841	
Readmitted (Class 1)	0.34	0.79	0.22	695	

Multi-class classifier

According to the comparison of multi-class classifiers, the following models achieved the higher AUC scores: random forest with oversampling (0.6368), random forest with class weight (0.6321), gradient boosting without any balancing and weighting technique (0.6319), and gradient boosting with oversampling (0.6306) (see Excel S3). However, random forest with oversampling had the highest AUC score and performed best across all classes on multiple evaluation metrics so it was selected as final multi-class classifier (see Table 6 below and Fig. S9).

Table 6 Result of optimal multi-class classifier.

Multi-class classifier	Random Forest with oversampling	
AUC	0.6368	
F1-Score (Test set)	0.46	
Accuracy (Test set)	0.46	
Classes	F1-Score	Recall	Precision	Support	
Short: within 6-months (Class 0)	0.46	0.46	0.46	198	
Medium: 6 months–2 years (Class 1)	0.40	0.38	0.43	264	
Long: greater than 2-years (Class 2)	0.51	0.55	0.48	224	

The dataset used in the final binary and multi-class classifiers contained 12 chosen features (see Table 3) and 200 columns of diagnoses and medications. These columns are the 100 most frequent diagnoses and medications, each represented using ICD-10 codes and level 5 ATC codes respectively. The final trained binary classifier (gradient boosting with oversampling) correctly identified 488 cases re-admitted and 2,348 cases not re-admitted. However, it incorrectly identified 1,493 cases as re-admitted and 207 cases as not-readmitted. On the other hand, the final trained multi-class classifier (random forest with oversampling) correctly identified 91 cases as having a short readmission period, 100 cases as having a medium readmission period, and 123 cases as having a long readmission period. However, it incorrectly identified 64 cases as medium, 43 cases as long when they were short; 73 cases as short, 91 cases as long when they were medium; and 33 cases as short and 68 cases as medium when they were long (see Confusion Matrix in Excel S2 and Excel S3).

Interpretation of classification results

This section provides an insight into how the classification models predicted the readmission classes of episodes.

The binary predictive model and the actual data both indicated that the majority of data points (12,777 and 19,250, respectively) were labeled as “not-readmitted” However, the model incorrectly predicted a higher number of “readmitted” labels (9,899) compared to the actual number (3,426) (see Excel S4). This discrepancy suggests that the predictive model was not very accurate in identifying the “readmitted” class (see Fig. 11).

Figure 11 Predicted binary class label vs actual binary bin label across “num_diagnoses”, “num_medication” and “gender”.

The multi-class classifier predicted 981 episodes labeled as “0–6 months”, 1,308 labeled as “6 months–2 years”, and 1,137 labeled as “over 2 years”. In comparison, the actual distribution was 1,001 episodes labeled as “0–6 months”, 1,316 labeled as “6 months–2 years”, and 1,109 labeled as “over 2 years” (see Excel S4). When the number of diagnoses and medications is low, the model often overestimates the readmission time, predicting more than 2 years instead of the actual 6 months–2 years. Conversely, for patients with more diagnoses or more medications, the model is more likely to predict shorter readmission times (see Fig. 12). Also based on the model, for males more diagnoses increase the chance of readmission in “6 months–2 years”, while more medications increase the chance of readmission in less than 6 months. For females, both more diagnoses and more medications lead to a higher probability of readmission in 6 months. The model is more accurate when the patients have more diagnoses and medications, as it follows the same patterns as the actual data.

Figure 12 Predicted multi-class label vs actual multi bin label across “num_diagnoses”, “num_medication”, and “gender”.

Across different age groups, the predicted and actual readmission rates seem more similar. However, the model still makes errors, such as for preschoolers with few diagnoses, as it predicts a longer readmission period than the actual data (see Fig. 13).

Figure 13 Predicted multi-class vs actual multi bin label across “num_diagnoses”, “num_medication” and “age_group”.

Additional insights can also be obtained (see Fig. S10). Teenagers typically have shorter episodes of care than other age groups but spend more time as inpatients. According to the model, the readmission time varies by age group. Preschoolers and middle childhood tend to be readmitted after 2 years, whereas teenagers have a higher likelihood of being readmitted within 2 years either within 0-6 months or 6 months to 2 years.

Clustering

Episodes of the patients were grouped together using clustering to determine the optimal number of clusters and evaluate their distinctiveness. K-prototype can effectively handle mixed data types, including numeric, categorical, and string/object types. All features in Table 3 and the target variable in Table 4 are used to compute the cluster labels. The Huang parameter, which is calculated using the occurrence frequency of the categorical attributes, was used to initialize the K-prototype clusters. A heuristic approach was used to find the optimal number of clusters. The elbow plot in Fig. 14 shows the elbow at five numbers of clusters, as there is a noticeable decrease in the cost function with respect to others.

Figure 14 Elbow plot with cost function (WCSS) vs number of clusters.

CI and SI scores are used to analyze the quality of clusters. Exclusively for numerical columns without diagnoses and medications data, Euclidean distance was used. For the rest, which required handling mixed data, we used the Gower distance (Gower, 1971). Gower measures the dissimilarity between two datasets with mixed data types. As Fig. 15 below shows, when using the dataset containing diagnoses and medications, an increase in the number of clusters leads to a decrease in both CI and SI scores. It indicates that the clustering algorithm struggles to find meaningful and well-separated groups with higher data dimensionality. The SI score in the dataset without diagnoses and medications shows an increase and decrease trend. In contrast, the CI score increases as the number of clusters in the dataset without diagnoses and medications increases, indicating that the clusters become more compact and well-separated. Choosing models with 20, 50, 100, or all diagnoses and medications may not be optimal. Better results might be achieved from a different set of diagnoses and medications.

Figure 15 SI and CI Score in different dataset at different number of clusters for the model selection.

As Fig. 15 above shows, without diagnoses and medications performed well overall, but it was not representative. The model that includes all diagnoses and medications has a good SI (well-matched own and poorly to nearby clusters) but a poor CI score (not dense and poorly separated). Based on the elbow’s cost function, we selected three (the immediate cluster after two) and five clusters (the elbow) for analysis and comparison (Fig. 14). Among the most frequent 20, 50, and 100 diagnoses and medications, the 20 was selected because it has comparatively higher average CI and SI scores. Comparing the most frequent 20 diagnoses and medications at three and five clusters, it shows that both SI and CI scores are higher in three clusters (SI: 0.256, CI: 4,473.64) than in five (SI: 0.118, CI: 2,997.435) indicating better-quality clusters. Below is a visual comparative analysis of the models with three and five clusters on the 20 most frequent diagnoses and medications (see CI and SI scores of all clustering models in Excel S5).

Cluster analysis

In the three cluster model with labels 0, 1, and 2, cluster 0 contained 16,818 episodes, cluster 1 had 4,522, and cluster 2 had 1,336. For the readmitted episodes (“Tillnextepisode” >0), Fig. 16 shows cluster 2 with the lowest range of “Tillnextepisode” had the highest range of “Length_of_Episode”, “Count_visit”, “Num_diagnoses”, and “Num_medications”. Conversely, Cluster 0 and 1 with the highest “Tillnextepisode” had comparatively lower “Length_of_Episode”, “Count_visit”, “Num_diagnoses”, and “Num_medications”, with varying ranges in other features. Figure 17 conveys similar information through a box plot.

Figure 16 Model with three clusters for readmitted episodes—Minima, maxima.

Figure 17 Model with three clusters for readmitted episodes—Box plot.

In the five cluster model, the episode distribution was: cluster 0 had 8,841 episodes, cluster 3 had 8,764, cluster 1 contained 3,711, cluster 2 had 1,223, and cluster 4 had the fewest with 137 episodes. For the readmitted episodes (“Tillnextepisode” >0), Fig. S11 shows that cluster 1, which has high values for “Length_of_Episode”, “Count_visit”, “Num_diagnoses”, and “Num_medications”, exhibits a low range for “Tillnextepisode”. Cluster 4 displays low values for “Length_of_Episode”, “Count_visit”, “Num_diagnoses”, “Num_medications”, and “TreatmentPlanning_ratio”, but a high range for “Tillnextepisode”. Cluster 3 shows a similar pattern to cluster 4 but with high “Therapy_ratio”, “TreatmentPlanning_ratio”, and “Advisory_ratio”. Clusters 0 and 2 have a moderate range for “Tillnextepisode” and varying ranges for the other features. Figure S12 also conveys similar information through a box plot. However, a long interquartile range (IQR) suggests that the middle 50% of values are widely dispersed. In contrast, a short IQR indicates values are closely packed together, showing low variability in the data. The dots with a circle show features where all values in the middle 50% are exactly the same, indicating no variation.

In both cluster five and cluster three models for readmitted episodes, the actual age (not age group) had almost similar maximum and minimum ranges, with slight variation (see Fig. 16 and Fig. S11). Also more males than females were observed, and no gender_0 episodes (see Figs. S17 and S18).

For readmitted and not-readmitted episodes, in both three and five cluster models, no discernible relationship between “Tillnextepisode_bins” and other features was observed (see Figs. S13–S16). Finally, analyzing the overall results of three and five clusters using CI, SI, minima, maxima, and box plots, we conclude that the three-cluster model is better.

Discussion and Future Work

To improve healthcare resource allocation, effective planning, and ensure the timely delivery of high-quality care within CAMHS, this study utilized machine learning techniques. The analysis of 35 years of clinical data captures the evolution of CAMHS organization and practices, offering insights for future improvements. However, several challenges posed significant difficulties in pre-processing and identifying relevant features for readmission prediction. These challenges included numerous inconsistencies and errors in the data, complexities in coding clinical states, and difficulties in understanding the process. Inputs from the clinicians and the reviewed literature guided the data pre-processing and feature selection, with specific guidance on the methodology provided by Matheny et al. (2021), Golas et al. (2018), Zeinalnezhad & Shishehchi (2024), and Betts, Kisely & Alati (2020). Despite extensive pre-processing and a readmission rate of 15.1%, the initial classifiers trained on the dataset struggled to predict certain classes due to significant imbalances. Specifically, the proportion of episodes without readmissions was 5.6 times higher than those with readmissions. In contrast to previous studies that primarily used single-method approaches, we cascaded the classification (binary and multi-class). The binary classifier acted as a filter, and the multi-class classifier provided a more detailed classification for the filtered subset.

We apply techniques such as weighting and oversampling to improve model predictions for readmission classes. Nonetheless, we were cautious in extensively using these techniques to avoid potentially distorting the quality of our data. For binary classification, the oversampled gradient boosting model, with an AUC of 0.7005 and an F1-score of 0.59, was the best performer overall. For multi-class classification, the oversampled random forest model achieved a slightly higher AUC of 0.6368 and an F1-score of 0.46, performing well in classifying readmissions across short, medium, and long classes. These results emphasize the challenge of predicting readmissions in CAMHS. However, our findings revealed certain connections between clinical features. For instance, we found that the number of diagnoses, medications, and visit patterns indicates that patients with more complex conditions and higher care intensity are more likely to be readmitted. This underscores the significance of considering case complexity and care intensity in clinical decision-making. Overall models showed a low predictability of readmission, this may possibly be due to several factors:

• Policy changes: in interviews with clinicians, we found that the policy was always not to discharge patients with severe problems, but sometimes, patients would be discharged and potentially readmitted elsewhere outside our cohort when clinicians changed jobs or workplaces. Such examples of patients being transferred among institutions can potentially distort the prediction of readmission.

• Finance and resource changes: Finances and resources at Norway’s CAMHS improved over time, and patient treatment at BUP St. Olavs Hospital rose from 420 in 1993 to 5,000 in 2018. This improvement and increase might have potentially influenced the results.

• Different municipalities, different resources, different discharge patterns: There are large differences in resources among the municipalities served by each CAMHS.

• Complexity in the nature of patient episodes of care: The complex nature of patient episodes in CAMHS, might lead to this unpredictability.

• Patients developing from children to adults: The conflict between discharging and retaining patients is particularly acute in CAMHS, as these patients are rapidly evolving, under development, and in a crucial phase of life.

• CAMHS effectively avoids inappropriate discharges: There is evidence that episodes involving patients with complex clinical concerns tend to be prolonged, often without a discernible discharge. The readmission pattern indicates that CAMHS effectively prevents inappropriate discharges that could lead to readmission, which is indirectly related to longer episodes. It may be that we should interpret some long episodes, with dips in activity, as equivalent to readmissions.

• Limited capacity for considering features: Many factors affect admission and readmission to CAMHS, we just had a small, limited view. Many aspects of data and processes were not included in this study. As an example, we excluded diagnoses on axis 4, 5, and 6 which encode changing psychosocial conditions, family situations, and CGAS, as these may influence the likelihood of the need for readmission. Further study is needed to evaluate if including this information about psychosocial situations, function, and family characteristics may improve the prediction of readmission.

These changes suggest that predicting readmissions in CAMHS is difficult and not easily learned by models. Despite this, we discovered cascading two classifiers useful, and observed some relationships between a few features, such as the number of diagnoses, number of medications, count of visits, standard deviation of monthly patient visits, and length of episodes, all of which would be indicators of care intensity and case complexity. Our findings demonstrate evidence of an association between intensity and complexity indicators and readmission. We hypothesize that patients with more complex conditions and intensity of care are more likely to return for later services.

Now that we know these complexities in readmission prediction, the challenging question is whether readmission is a problem. If it is, it is important to identify the groups for whom it is a significant problem. Knowing those patients is critical for improving healthcare services. Another question is whether there is a need to prevent readmissions. Is it bad or good? Hospitals may see it as bad because it uses resources. But what about physicians, patients, and families? Could they see it differently? Another question is how to prevent it. A simple solution to prevent readmissions would be to stop discharging patients. However, this strategy is impractical because it would saturate services and leave no room for new patients. So it seems it is important to identify the patients who should continue to receive services as they are likely to be the patients who cannot manage their lives without specialist services. This group usually includes the sickest patients and those with limited family support. And the next question is what the data suggest and conclude. Our findings showed challenges in predictability, but further research is necessary to explore different methods for understanding what our data can teach us. We need to interpret more of the available textual narrative to understand progress, disposition, potential, and treatment options. A better understanding of clinical practice outcomes and data analysis results in CAMHS is essential for taking action to improve care.

Contribution

Through this comprehensive analysis of CAMHS data, we identified key characteristics related to readmission prediction. Although predicting readmissions using selected features from EHR data proved challenging, the approach of cascading binary and multi-class classifiers demonstrated utility in improving prediction performance. Our most optimal models (oversampled Gradient Boosting and Random Forest) ensure good performance across all classes, particularly emphasizing the minority class. For clustering, three was the optimal number of clusters, revealing some patterns between features such as length of episode, number of diagnoses, and medication with readmission. Our research employs predictive models that focus on care intensity and case complexity. This approach, distinct from other studies, deepens the understanding of readmission risks specific to CAMHS. By emphasizing care intensity and case complexity in predictions, our findings can contribute to improving patient management and reducing readmission rates.

Limitations

A potential limitation of our approach is that we treat readmission as a stationary phenomenon across the CAMH service over many years, and randomly selected shuffled training and test data from the entire time span of our dataset. An alternative approach would be to select the training set from a range of earlier years, ensuring that test data comes from the later time period, under the assumption that prior readmission practice developed into and informed later practice. However, all these results could have been explored in more detail, but according to CAMHS management, readmission practice has been stable for the patient group in question, thus our results may be optimistic, but not invalid.

Also, ML methods can find predictors correlated with outcomes, they do not inherently uncover causal relationships. To get better scientific understanding, we need predictors causally related with outcome. Lastly we focused on structured data, a more comprehensive analysis incorporating unstructured clinical notes could yield deeper insights into readmission factors, potentially improving model performance and understanding.

Future work

The generalizability of our findings depends on the specifics of CAMHS care services. We believe our results are applicable to Norway and other Nordic countries. We plan to validate this further with additional features, through multi-cohort and multi-hospital studies. However, differences in healthcare system organization, such as varying referral periods and continuity of responsibility, may limit their applicability to other regions. Despite this, our findings could still offer potential benefits within distinct service organizations. To improve readmission predictions, future studies should include psychosocial factors, family dynamics, social context, patient development, disease progression, and intervention effects for a holistic view of patient care. Further research should explore various machine learning methods, including causal inference techniques, hyper-parameter optimization, and diverse classifiers. Additionally, using different internal validation metrics for clustering is recommended.

Supplemental Information

Supplemental Information 1 Count of diagnoses in each axis

Supplemental Information 2 Most frequent diagnosis (ICD-10) codes with their counts

Supplemental Information 3 Most frequent medication codes with their counts

Supplemental Information 4 Distribution across key categories

Supplemental Information 5 Overview of admissions and discharges (1987-01-01 : 2019-07-03)

Supplemental Information 6 The initial feature correlation matrix

Supplemental Information 7 Scatter plot of the features and target variable ”Tillnextepisode”

Supplemental Information 8 ROC curves for various binary classification algorithms with different balancing and weighting techniques

Supplemental Information 9 ROC curves for various multi-class classification algorithms with different balancing and weighting techniques

Supplemental Information 10 Predicted multi-class label vs actual multi bin label across ”Length_of_Episode”, ”Inpatient_ratio” and ”age_group”

Supplemental Information 11 Five cluster model for readmitted episodes - Minima, maxima

Supplemental Information 12 Five cluster model for readmitted episodes - Box plot

Supplemental Information 13 Five cluster model using readmitted and not-readmitted episodes - Minima, maxima

Supplemental Information 14 Five cluster model using readmitted and not-readmitted episodes - Box plot

Supplemental Information 15 Three cluster model using readmitted and not-readmitted episodes - Minima, maxima

Supplemental Information 16 Three cluster model using readmitted and not-readmitted episodes - Box plot

Supplemental Information 17 Three cluster model for readmitted episodes - Gender distribution

Supplemental Information 18 Five cluster model for readmitted episodes - Gender distribution

Supplemental Information 19 Comparing the classifiers for different numbers of frequent diagnoses and medications, Phecodes and ATC4

Supplemental Information 20 Comparing the binary classifiers for different classes with different metrics

Supplemental Information 21 Comparing the multi-class classifiers for different classes with different metrics

Supplemental Information 22 Comparing the class distributions in different classifiers using various methods to handle imbalance data

Supplemental Information 23 CI, SI scores, and distribution of episodes in all clustering models

Supplemental Information 24 Hyper-parameters

Additional Information and Declarations

Competing Interests

Author Contributions

Ethics

Data Availability

The authors declare there are no competing interests.

Kaban Koochakpour conceived and designed the experiments, performed the experiments, analyzed the data, performed the computation work, prepared figures and/or tables, authored or reviewed drafts of the article, data pre-processing and data preparation, and approved the final draft.

Dipendra Pant performed the experiments, analyzed the data, performed the computation work, prepared figures and/or tables, authored or reviewed drafts of the article, and approved the final draft.

Odd Sverre Westbye analyzed the data, authored or reviewed drafts of the article, clinical interpretation of data and results, and approved the final draft.

Thomas Brox Røst analyzed the data, authored or reviewed drafts of the article, verification of Technical Implementation, and approved the final draft.

Bennett Leventhal analyzed the data, authored or reviewed drafts of the article, clinical interpretation of data and results, and approved the final draft.

Roman Koposov analyzed the data, authored or reviewed drafts of the article, clinical interpretation of data and results, and approved the final draft.

Carolyn Clausen analyzed the data, authored or reviewed drafts of the article, and approved the final draft.

Norbert Skokauskas analyzed the data, authored or reviewed drafts of the article, clinical interpretation of data and results, and approved the final draft.

Øystein Nytrø conceived and designed the experiments, analyzed the data, authored or reviewed drafts of the article, verification of Technical Implementation, and approved the final draft.

The following information was supplied relating to ethical approvals (i.e., approving body and any reference numbers):

The Regional Committee for Medical and Health Research Ethics (REK), under reference number 15600, South East assessed the IDDEAS application and concluded that “the project falls outside the scope of the Health Research Act, cf. section 2, and can therefore be implemented without approval of REK” (reference number: 2018–2186, 09/10/2019). As per recommendation of REK, the IDDEAS research team submitted a risk analysis to Central Norway Regional Health Authority IT and a data protection impact assessment (DPIA) to the local data protection officer, and approval was received.

The following information was supplied regarding data availability:

The code is available at Zenodo: Koochakpour, K., Pant, D., Nytrø, Ø., Odd Sverre, W., Thomas Brox, R., Bennett L., L., Roman, K., Carolyn, C., & Norbert, S. (2024). Ability of clinical data to predict readmission in Child and Adolescent Mental Health Services (Version V0). Zenodo. https://doi.org/10.5281/zenodo.12534475.

This article used the BUPdata electronic health records managed by St. Olavs Hospital in Trondheim, Norway. Due to the sensitive nature of the data, it cannot be made publicly available with the article.

To request access, please send an email to forskningsavdelingen@stolav.no. The applicant must comply with any requirements or guidelines provided by the head of the project, non-disclosure agreements, REC, DAC, and HUNT Cloud. Breach of legal or formal requirements is punishable under Norwegian law.

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
