# Peer review of "Ability of clinical data to predict readmission in Child and Adolescent Mental Health Services"

_PeerJ Computer Science, doi:10.7717/peerj-cs.2367_

## Round 0.1 · original submission · Major Revisions

We had to solicit over 20 reviewers in order to finally obtain four reviews. Unfortunately, none of the reviews are of high quality. Two are excessively vague and look like they were written by AI.

None of the reviews identified the worst weakness of this submission, which is that it does not explain whether the reported accuracy numbers are on an independent test set. Line 491 mentions cross-validation, with no further discussion anywhere. The revised submission must explain this fully and carefully.

Note that accuracy numbers measured by cross-validation are still likely to be over-optimistic. A more realistic procedure is to make the test data be later in time than the training data. Readmission is likely to be a non-stationary phenomenon, where the factors correlated with readmission change over time.

Another topic that the paper must discuss is correlation versus causation. ML methods merely find predictors correlated with the outcome. To get useful scientific understanding, we need predictors to be causally related with the outcome.

The paper must also be explicit about which statistical tests are used. Line 478 says "The results did not show a significant difference." Based on what statistical test? Note that when comparing different classifiers on the same test set, the correct test to use is likely to be the McNemar test; see https://rasbt.github.io/mlxtend/user_guide/evaluate/mcnemar/

More generally, the authors should follow a checklist for applying supervised learning in medicine. See https://www.sciencedirect.com/science/article/pii/S1386505621001362?via%3Dihub and https://insightsimaging.springeropen.com/articles/10.1186/s13244-022-01355-9 for alternatives.

Reviewer 1 ·

Basic reporting

no comment

Experimental design

The manuscript offers an intriguing exploration within the Aims and Scope of the journal, addressing the significant issue of readmissions in Child and Adolescent Mental Health Services (CAMHS). However, the research question could be articulated more clearly to explicitly state how this study fills a specific knowledge gap within the field. Enhancing the introduction with a more distinct statement of the research question and its relevance would greatly benefit the manuscript's alignment with the journal's objectives and illuminate its contribution to the field.

While the investigation appears thorough, ensuring the highest standards of technical and ethical rigor requires a more detailed exposition of the methodologies employed, including data collection, analysis techniques, and ethical considerations. Elaborating on these aspects will not only bolster the credibility and reliability of the findings but also adhere to the scholarly standards expected by the journal.

Regarding the methods section, it is recommended to provide a more comprehensive description of the procedures and techniques used in the study. This includes detailing the data preprocessing steps, the selection criteria for the machine learning models, and any validation techniques employed. Expanding on these details will enhance the reproducibility of the study, a crucial aspect of scientific research. Providing such depth will enable peers to replicate the study, thereby validating its findings and contributing to the robustness of the research within the CAMHS domain.

Validity of the findings

The manuscript provides intriguing findings that contribute to the understanding of readmission patterns in Child and Adolescent Mental Health Services (CAMHS). However, the impact and novelty of these findings could be more thoroughly assessed to clearly delineate the study's contribution to the existing literature. It would be beneficial for the authors to explicitly discuss how their work advances our understanding or introduces new perspectives on the topic. Encouraging replication studies, with a clear rationale and potential benefits to the field, could further underscore the significance of the findings.

Reviewer 2 ·

Basic reporting

All comments have been added in detail to the 4th section called additional comments.

Experimental design

All comments have been added in detail to the 4th section called additional comments.

Validity of the findings

All comments have been added in detail to the 4th section called additional comments.

Additional comments

Review Report for PeerJ Computer Science
(Ability of clinical data to predict readmission in child and adolescent mental health services)

1. Within the scope of the study, analysis and clustering of CAMHS services in different periods was carried out.

2. A large number of care episodes and patients over many years were used as the dataset. The dataset, which contains many different data structures such as interventions and diagnoses, has been subjected to various pre-processes instead of being used raw. This stage is at an acceptable level and the preprocessing used is sufficient.

3. In order to understand the literature review given before the Data and Materials section more clearly and to emphasize the importance of the problem; It is recommended to add a literature comparison table consisting of columns such as "method, pros, cons, results". In addition, immediately after this, it should be recommended that the study be explained in more detail by making comparisons with the literature in bullet points in order to explain more clearly the originality of the study, its contribution to the literature and its difference from the literature.

4. When the study was examined in terms of method, it was observed that multi-class classification and clustering were performed in addition to binary classification. It is stated that XGBClassifier is used for binary classification and Logistic Regression is used for multi-class classification. When the literature is examined in detail, it is known that there are a wide variety of machine learning models that can be used for the classification stage of problem solving. However, a very limited number of classifiers were used in this study. The reason for not using other models and/or choosing these models in particular should be explained in detail and what contribution this makes to originality should be explained.

5. It is very important to analyze the classification results correctly and obtain evaluation metrics accurately and completely. For this reason, it is recommended to obtain all missing metrics such as ROC curve and AUC score in this study.

As a result, although the study is of a certain quality, it is recommended to pay attention to the parts mentioned above.

·

Basic reporting

Thank you so much for this important study on readmission prediction in children and adolescents with psychiatric issues. It led to a number of interesting results that is challenging to predict readmissions. Overall paper used clear English throughout with some exceptions listed below.

All Figures had no explanations, and I had a hard time orienting myself with them.
Tables also could use some legends.

What does CAMHS stand for? It might have been mentioned in the paper but I had a hard time finding it?
Methods: clarify 6 months to 2 years. Right now it reads as 6 to 2 years.

Line 353 seems like a type: IDDEAS of which this study is a part of that…

Discussion: first statement oddly phrased and does not summarize the results.
Line 630 in discussion is oddly phrased.

Policy changes under discussion: is this true that patients would simply get discharged bc physician left a job?

Experimental design

Overall, I have no problems with the experimental design of the study. I am wondering if authors could explain a little more why they excluded psychosocial information from the axial diagnosis system. I believe that it is likely a very important reason contributing to readmission and a great limitation of the study.

Validity of the findings

All underlying data have been provided and are statistically sound. I do think however that discussion could be strengthened. Authors mostly use anecdotal information and not support their ideas with studies. Introduction has a nice scoping review and shows that authors did a lit search in preparation for their paper, that depth of analysis is not evidenced in their discussion. It is also customary to clearly outline limitations of study.

Additional comments

NA

Reviewer 4 ·

Basic reporting

The bibliographic information of references is incomplete.

You should expand upon the knowledge gap being addressed by your study.

The numbering of tables and figures in your text and supplementary materials does not match.the labels. For example, there is no Table 4-1 or 4-2

There are inconsistency in naming figures/tables for example you have used plot1-1 and Figure 13 for the same figure
.
Plot 1-1, plot 2-1 should be redesigned.

You may want to consider using a different image for Figures 3 and 4 to improve their visual presentation.

Figures 24, 25, and 26 are not readable.

Figures 22 and 23 should have higher quality.

You have the SQL file but not the raw data.

Experimental design

The reasons for different exclusions are not clear on page 11.


The methods are not described with sufficient detail and information to replicate the study. You haven't discussed parameter setting or hyperparameter tuning in your classification algorithms

How have you used Euclidean distance for mixed data?

Validity of the findings

Impact and novelty have not been assessed properly.

Raw data is not provided

The overall accuracy and F1-score of the logistic regression model are low.You may need to other methods or apply hyper parameter tuning


It is unclear why and how you have used clustering.

Additional comments

To improve the overall quality of the paper, I recommend focusing on enhancing the logical flow of the document. Here are some suggestions:

Clearly outline the objectives and research questions at the beginning.and explicitly state how your study addresses the identified knowledge gap.

Provide a detailed explanation of the methods and parameter used.Each step should logically follows from the previous one

Ensure that all figures and tables are correctly numbered and referenced in the text a

---

## Round 0.2 · accepted · Accept

For efficiency, this submission can be accepted and published now. Thank you to the authors for considering carefully the previous feedback, and for addressing it.

Here are some suggestions that I hope the authors will follow in the final version:

Please include the McNemar test results, with a brief discussion. Please also cite the ML checklist that you followed, saying that you followed it.

Currently the manuscript has Discussion and Conclusion sections on lines 582 to 689. It is common in machine learning to combine these but to organize them into parts that address (a) contributions, (b) limitations, and (c) future work.

In the limitations part, mention explicitly that accuracy numbers measured by cross-validation are likely to be over-optimistic; a more realistic procedure is to make the test data be later in time than the training data. Readmission is likely to be a non-stationary phenomenon, where the factors correlated with readmission change over time.

Also mention that ML methods merely find predictors correlated with the outcome. To get useful scientific understanding, we need predictors to be causally related with the outcome.

Separately, this paper has a long structured abstract, from lines 23 to 64. That is common in medical journals, but AFAIK, PeerJ expects an abstract that is a single paragraph with no specific organization.

Reviewer 2 ·

Basic reporting

All comments have been added in detail to the last section.

Experimental design

All comments have been added in detail to the last section.

Validity of the findings

All comments have been added in detail to the last section.

Additional comments

Thanks for the revision. The changes made to the paper and the answers given are generally at a certain level.

·

Basic reporting

See below

Experimental design

See below

Validity of the findings

See below

Additional comments

I believe authors answered my concerns satisfactorily.